# Haplotype function score improves biological interpretation and cross-ancestry polygenic prediction of human complex traits

**Weichen Song[1,2]\*, Yongyong Shi[2,3]\*, Guan Ning Lin[1]\***

[1]Shanghai Mental Health Center, Shanghai Jiao Tong University School of Medicine, School of Bioengineering, Shanghai Jiao Tong University, Shanghai, China; [2]Bio-X Institutes, Key Laboratory for the Genetics of Developmental and Neuropsychiatric Disorders (Ministry of Education), Collaborative Innovation Center for Brain Science, Shanghai Jiao Tong University, Shanghai, China; [3]Biomedical Sciences Institute of Qingdao University (Qingdao Branch of SJTU Bio-X12 Institutes), Qingdao University, Qingdao, China

**\*For correspondence:**
song628196@gmail.com (WS);
shiyongyong@gmail.com (YS);
nickgnlin@sjtu.edu.cn (GNL)

**Competing interest:** The authors declare that no competing interests exist.

**Abstract** We propose a new framework for human genetic association studies: at each locus, a deep learning model (in this study, Sei) is used to calculate the functional genomic activity score for two haplotypes per individual. This score, defined as the Haplotype Function Score (HFS), replaces the original genotype in association studies. Applying the HFS framework to 14 complex traits in the UK Biobank, we identified 3619 independent HFS–trait associations with a significance of p < $5 \times 10^{-8}$. Fine-mapping revealed 2699 causal associations, corresponding to a median increase of 63 causal findings per trait compared with single-nucleotide polymorphism (SNP)-based analysis. HFS-based enrichment analysis uncovered 727 pathway–trait associations and 153 tissue–trait associations with strong biological interpretability, including 'circadian pathway-chronotype' and 'arachidonic acid-intelligence'. Lastly, we applied least absolute shrinkage and selection operator (LASSO) regression to integrate HFS prediction score with SNP-based polygenic risk scores, which showed an improvement of 16.1–39.8% in cross-ancestry polygenic prediction. We concluded that HFS is a promising strategy for understanding the genetic basis of human complex traits.

## eLife assessment

This **valuable** paper presents a new approach for association testing, using the output of neural networks that have been trained to predict functional changes from DNA sequences. As such, the approach is an interesting addition to statistical genetics, and the evidence for the presented method being able to identify trait-associations in regions where GWASs are typically underpowered is **solid**. A limitation is, however, that it is unclear how the quality of these associations compares to those detected using conventional methods. Additional work assessing this method's power and characterizing false positives / false negative regions would be critical to ensure that the method is broadly adopted by the field.

## Introduction

Genome-wide association studies (GWAS) have witnessed remarkable advancements over recent years, both in terms of sample size and genetic discovery. However, the elucidation of downstream

**eLife digest** Scattered throughout the human genome are variations in the genetic code that make individuals more or less likely to develop certain traits. To identify these variants, scientists carry out Genome-wide association studies (GWAS) which compare the DNA variants of large groups of people with and without the trait of interest.

This method has been able to find the underlying genes for many human diseases, but it has limitations. For instance, some variations are linked together due to where they are positioned within DNA, which can result in GWAS falsely reporting associations between genetic variants and traits. This phenomenon, known as linkage equilibrium, can be avoided by analyzing functional genomics which looks at the multiple ways a gene's activity can be influenced by a variation. For instance, how the gene is copied and decoded in to proteins and RNA molecules, and the rate at which these products are generated.

Researchers can now use an artificial intelligence technique called deep learning to generate functional genomic data from a particular DNA sequence. Here, Song et al. used one of these deep learning models to calculate the functional genomics of haplotypes, groups of genetic variants inherited from one parent. The approach was applied to DNA samples from over 350 thousand individuals included in the UK BioBank. An activity score, defined as the haplotype function score (or HFS for short), was calculated for at least two haplotypes per individual, and then compared to various complex traits like height or bone density.

Song et al. found that the HFS framework was better at finding links between genes and specific traits than existing methods. It also provided more information on the biology that may be underpinning these outcomes. Although more work is needed to reduce the computer processing times required to calculate the HFS, Song et al. believe that their new method has the potential to improve the way researchers identify links between genes and human traits.

mechanisms and subsequent applications still face certain limitations (*Visscher et al., 2017*). One caveat is that the statistical power of GWAS on a variant relies on its population frequency (*Li et al., 2020*; *Null et al., 2022*; *Zhou et al., 2022*), whereas most variants with large effect size are rare (*Zeng et al., 2021*), leading to insufficient discoveries. Moreover, linkage disequilibrium (LD) among neighboring variants can significantly inflate false positive results (*Nowbandegani et al., 2022*). The variability of LD structure among different populations further compounds the challenges associated with training predictive models and discovering causal genes. Lastly, most trait-relevant variants reside in non-coding regions (*Watanabe et al., 2019*), which lack direct functional annotations as coding variants. The prevalent approach to addressing this issue is to annotate each variant based on its location within functionally significant regions (*Finucane et al., 2015*; *Grotzinger et al., 2022*; *Iotchkova et al., 2019*; *Weissbrod et al., 2020*; *Zheng et al., 2022*), such as transcription factor-binding sites or enhancers. While this strategy has considerably advanced the analysis, it is not optimal, as a variant's placement within a functionally important region does not inherently signify that the variant has substantial functional impacts.

The central dogma, proposing that DNA alterations' effects on phenotype are mediated via RNA and protein changes, offers a novel strategy to address these challenges. More precisely, by replacing the original genotypes in association studies with the aggregated impact of variants on transcription or functional genomics, the central dogma ensures the preservation of the majority of genetic information. This 'aggregated impact' offers several benefits for GWAS analysis: it provides direct biological interpretations, bypasses the effects of LD and population genetic history, and amalgamates information from both common and rare variants. One successful implementation of this strategy is Polygenic Transcriptome Risk Scores (PTRS) (*Hu et al., 2022*; *Liang et al., 2022*), which employ genetically determined transcription levels rather than genotypes to predict complex trait, and achieved remarkable portability. Nonetheless, the accuracy of imputing transcription levels from genotypes, given the sample size of currently available cohorts such as the Genotype-Tissue Expression project, GTEx (*Aguet et al., 2020*), remains limited ($R^2$ around 0.1 for most genes) (*Barbeira et al., 2018*). Thus, the performance of PTRS is yet to reach its optimal potential.

Following the success of PTRS, we made one step forward to utilize functional genomics in this strategy. Compared with transcription levels, predicting genetically determined functional genomic levels has achieved much higher accuracy by multiple recent deep learning (DL) studies (*Avsec et al., 2021*; *Chen et al., 2022*; *Kelley, 2020*; *Yan et al., 2021*; *Zhou et al., 2018*). These DL models utilize segments of the human reference genome as training samples, substantially increasing the sample size. Furthermore, functional genomics serve as a mediator between DNA and transcription, thus lessening the influence of non-genic factors such as the environment. Given these advancements, we propose that using the outputs of one of the state-of-the-art DL models, Sei (*Chen et al., 2022*), as the 'aggregated impact' in this novel strategy could effectively address the challenges aforementioned. Sei accepts a DNA sequence and computes multiple sequence class scores that represent different facets of the functional genomic activities of that sequence. This score integrates impacts from all variants, even those as rare as singletons, into one continuous variable, and is, in theory, unaffected by LD. In line with this notion, a recent similar strategy called cistrome-wide association study integrated variant–chromatin activity and variant–phenotype association to boost power of genetic study of cancer (*Baca et al., 2022*).

In this study, we present an analytical framework founded on this strategy (*Figure 1*) and implement it on complex traits in the UK Biobank to pinpoint causal loci and genes, decipher biological mechanisms, and devise cross-ancestry prediction models. We segmented the human reference genome into multiple 4096 bp loci, generated DNA sequences for each locus for two haplotypes per individual, and employed Sei to compute the functional genomic activities of these sequences. We designated this activity score as the Haplotype Function Score (HFS) and analyzed the association between the HFS and each trait. Our findings confirm that the HFS framework offers a unique improvement in the biological interpretation and polygenic prediction of complex traits compared to classic SNP-based methods, thereby demonstrating its value in genetic association studies.

## Results

### Overview of genome-wide HFS

We used the HFS framework to analyze imputed genotype data from the UK Biobank (*Figure 1*). We segmented the human genome (hg38) into 617,378 discrete, non-overlapping loci, each 4096 base pairs long. Of these, 590,959 loci carried at least one non-reference haplotype in the UKB cohort (see Method and *Supplementary file 1a*). After quality control, these loci contained approximately 1.2 billion haplotypes, with a median count of 819 per loci (*Figure 1—figure supplement 1*). We then employed the DL framework, Sei (*Chen et al., 2022*), to compute sequence class scores for each haplotype. In its sequence mode, Sei accepts DNA sequences in fasta format and produces multiple distinct sequence class scores, 39 of which were included in our study (Method). Our analysis identified significant variation in sequence class scores across different loci. In fact, 49.7% of loci housed haplotypes whose sequence class (as defined by the maximum of the 39 sequence class scores) differed from the reference haplotype sequence class. Using the reference sequence class as a benchmark, we noted that 16.8% of loci showed a difference between the maximum and minimum haplotype scores that surpassed the score of the reference haplotype. Moreover, the correlation between sequence class scores of adjacent loci was low, with a median $R^2$ value of 0.013 (*Figure 1—figure supplement 2*), effectively reducing the impact of LD in association studies. Further evaluation indicated that this low LD was led by two factors: integration of rare variant impacts and segmentation. Firstly, excluding rare variants from HFS caused the LD raised to median = 0.14 (Method; *Figure 1—figure supplement 2C*). Secondly, median LD of SNPs from adjacent loci was 0.06, which was significantly higher than HFS LD (paired Wilcoxon p = $1.76 \times 10^{-5}$) but significantly lower than HFS LD without rare variants (paired Wilcoxon p < $2.2 \times 10^{-16}$).

Expanding on the sequence class scores, we defined HFS for each locus. Specifically, we computed the mean sequence class score of two haplotypes per individual, reflecting an additive model. We selected the score corresponded to the sequence class of reference sequence as the HFS of the corresponding locus, and its association with each trait was computed using a generalized linear model. Simulation analysis revealed that when a non-reference sequence class score was associated the trait, reference class score could still capture median 70% of HFS–trait association $R^2$. We applied this framework to 14 polygenic traits in the UKB British ancestry training set (n = 350,587; *Supplementary*

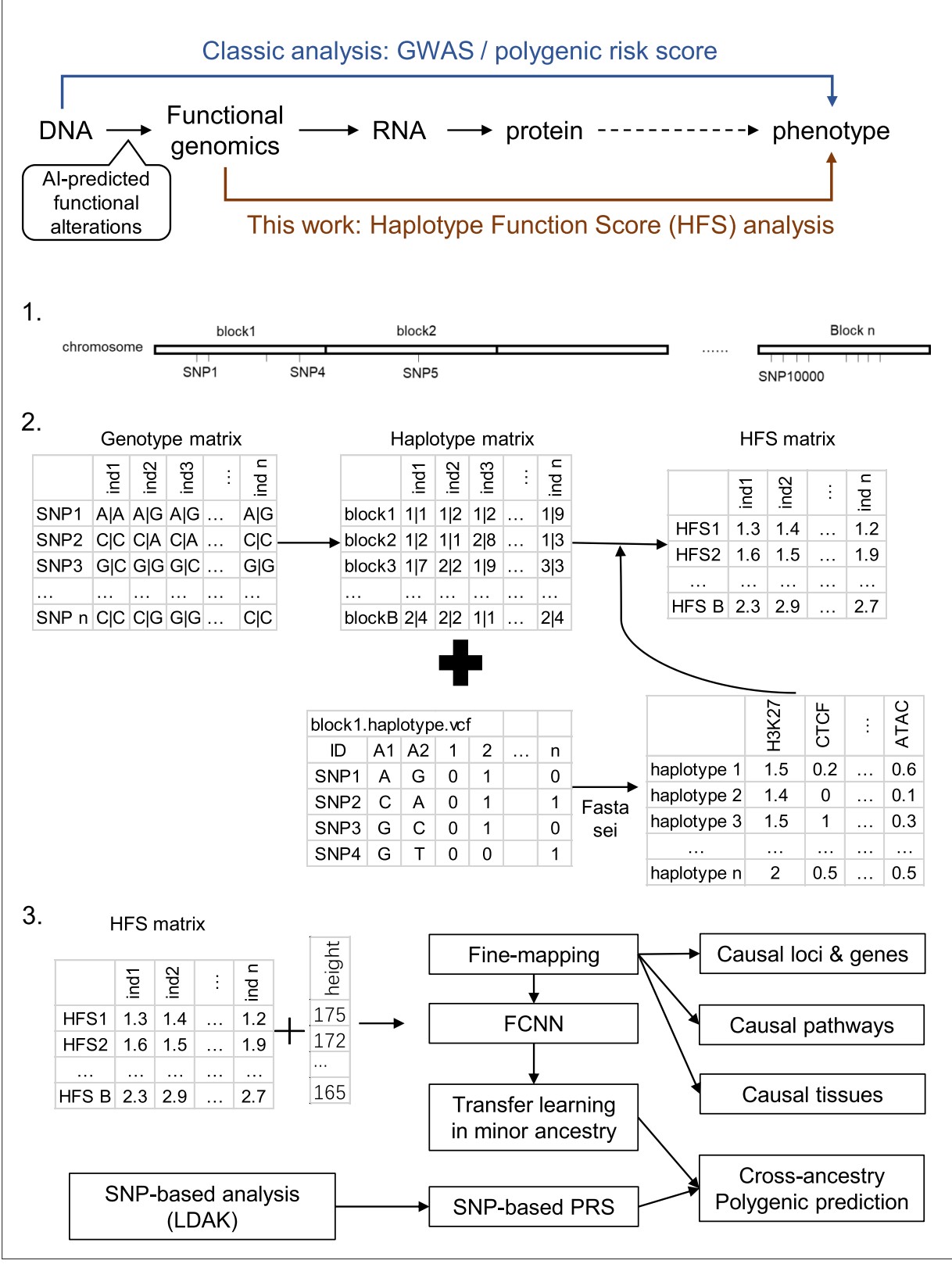

**Figure 1.** Flowchart of the study. Ind: individual.

The online version of this article includes the following figure supplement(s) for figure 1:

**Figure supplement 1.** Distribution of number of haplotypes per locus.

*Figure 1 continued on next page*

*Figure 1 continued*

**Figure supplement 2.** Linkage disequilibrium (LD) among Haplotype Function Score (HFS).

**Figure supplement 3.** Comparison of inflation factor between Haplotype Function Score (HFS) and SNP association tests.

*file 1b* and Method), identifying 16,597 significant HFS–trait associations at a threshold of p < 5 × $10^{-8}$ (n = 15 for insomnia, n = 7573 for height; *Supplementary file 1b*), equating to roughly 3619 independent associations. The most significant associations were between the 'promotor' score of chr7:121327898–121331994 (WNT16) and bone mineral density (BMD; regression beta = −0.02, p < $10^{-300}$), and the 'promotor' score of chr9:4760952–4765048 (AK3) and platelet count (beta = 3.20, p = 2.79 × $10^{-262}$; *Supplementary file 1c*).

When comparing HFS association with the standard SNP-based GWAS on the same data, we found that 98% of significant HFS loci also harbored a significant SNP. There were a few cases (n = 0–5) where significant HFS loci did not harbored even marginal SNP association (GWAS p > 0.01), which were due to the lack of common SNP in these loci. HFS association p-value was higher than GWAS p-value in 95% of significant loci, suggested that HFS did not improve power to detect marginal effect. The genomic control inflation factor ($\lambda_{GC}$) for the HFS association test varied between 0.99 for asthma and 1.50 for height, closely resembling the SNP GWAS (Pearson correlation coefficient [PCC] = 0.91, paired *t*-test p = 0.16; Method and *Figure 1—figure supplement 3*). We concluded that HFS-based association tests had adequate power and do not introduce additional p-value inflation.

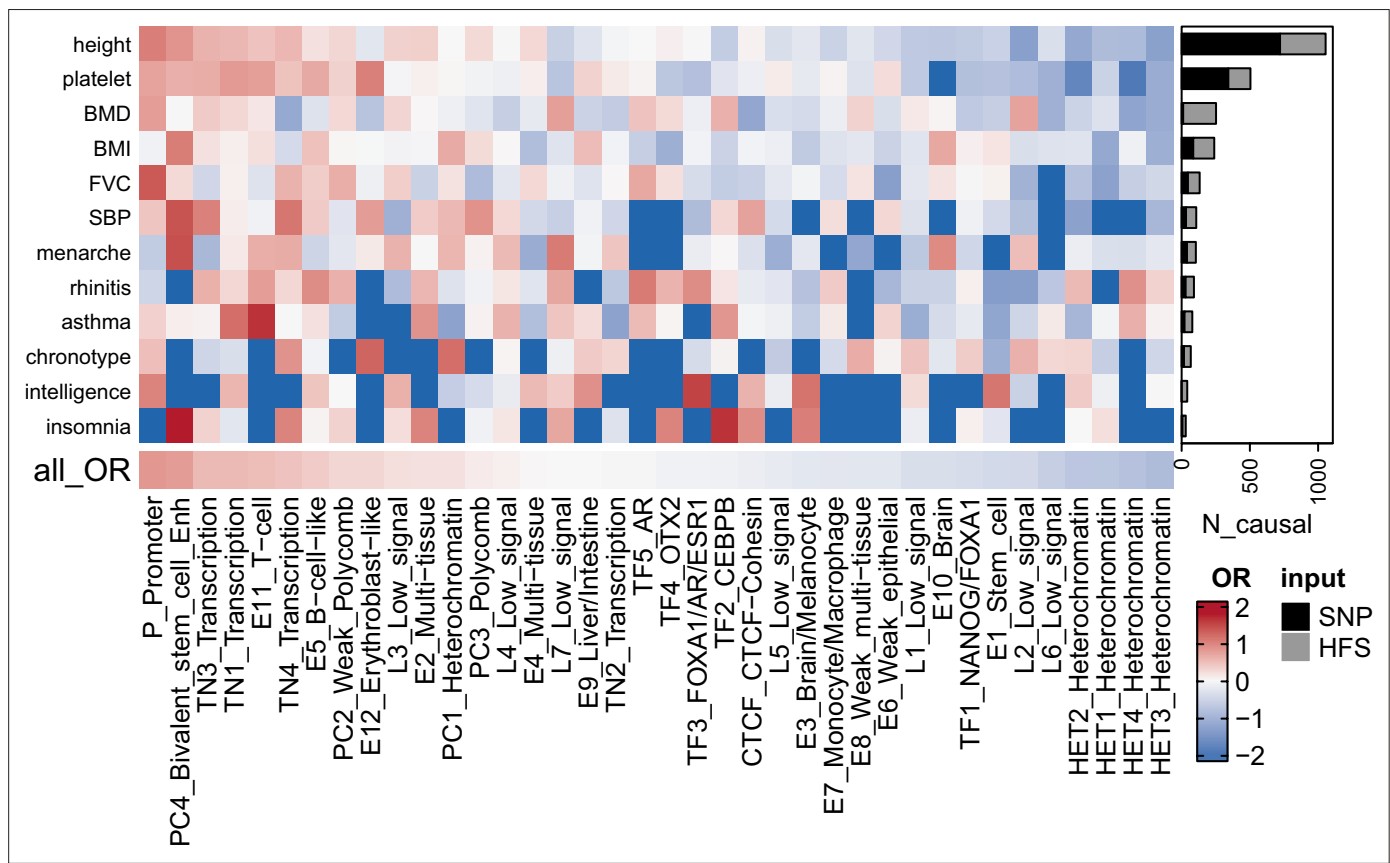

**Figure 2.** Fine-mapping result summary. Gray bar plots indicated the number of loci with posterior inclusion probability (PIP) >0.95 in Haplotype Function Score (HFS) + SUSIE (causal loci). Black bar plots indicated number of SNP with PIP >0.95 in PolyFun or SbayesRC analysis (the larger number was shown). Each grid of heatmap showed the odds ratio of each sequence class loci being causal loci for each trait. 'All_OR' indicated odds ratio for pooling all traits together. Enh: enhancer. TF: transcription factor-binding site.

The online version of this article includes the following figure supplement(s) for figure 2:

**Figure supplement 1.** Heritability enrichment within causal loci estimated Linkage Disequilibrium Score regression (LDSC).

## Fine-mapping based on HFS

Based on these data, we applied SUSIE to fine-map the causal loci that were associated with each of the 14 traits. We divided hg38 genome into 1361 independent blocks as defined by *MacDonald et al., 2022*, and applied SUSIE (*Wang et al., 2020*) to loci HFS in each of these blocks (number of loci per block = 4–2392). As shown in *Figure 2* and *Supplementary file 1d*, we identified a total of 2699 causal loci–trait associations at the threshold of posterior inclusion probability (PIP) >0.95, hereafter referred to as 'causal loci'. Compared with SNP-based functionally aware fine-mapping methods PolyFun (*Weissbrod et al., 2020*) and SbayesRC (*Zheng et al., 2022*), HFS-based SUSIE detected −11 to 334 more causal signals (median = 63, *Supplementary file 1e*) for each trait. We cautioned that these methods use summary statistics as input and are by nature less sensitive than individual data-based methods. Yet, we suggested that such impact would be mild, since we used in-sample LD reference (from UKB European sample).

Among these causal loci, only 22% were also lead loci in association analysis (loci with the lowest p-value in 200 kb region), and 58% had association p-value >$5 \times 10^{-8}$. In line with previous SNP-based analysis (*Weissbrod et al., 2020*), this result highlighted the importance of using causal signals instead of lead signals in post-GWAS analysis. We found 67 causal loci showing pleiotropic effects on at least two independent traits, including 'CTCF-Cohesin' score of chr9:89596537–89600633 that was associated with age at menarche, body mass index (BMI) and height (PIP >0.97; *Supplementary file 1d*). We also found that rare variants played an important role in the good find-mapping performance of HFS: when variants with MAF <0.01 were removed, 55.3% of the causal signals would be missed in HFS + SUSIE analysis.

When looking at the reference sequence class of loci, those with functional importance were more likely to be causal loci, including 'Promoter' (odds ratio [OR] = 2.33, p = $1.41 \times 10^{-14}$), 'Bivalent stem cell enhancer' (OR = 2.22, p = $1.11 \times 10^{-8}$), and 'Transcribed region 1' (OR = 1.71, p = $1.581 \times 10^{-10}$, *Figure 2*). Such functional enrichment was even higher for pleiotropic loci ('Promoter': OR = 7.20, p = $3.35 \times 10^{-5}$). We also observed trait-specific patterns of such sequence class enrichment, such as 'CEBPB-binding site' (Insomnia: OR = 5.25, p = 0.01) and 'FOXA1/AR/ESR1-binding site' (intelligence: OR = 4.69, p = 0.01, *Figure 2* and *Supplementary file 1f*). These results demonstrated the expected functional patterns of causal loci, and indicated that HFS-based fine-mapping was biologically interpretable and reliable.

Despite the functional enrichment, we applied several secondary analyses to verify the reliability of HFS-based SUSIE result. Firstly, we took causal SNP fine-mapped by PolyFun (*Weissbrod et al., 2020*) as positive control, and find that compared with genomic region-matched control loci, causal loci were significantly enriched for causal SNP (OR = 1.33–5.08, Fisher's test p = 0.12–$4.72 \times 10^{-52}$, *Supplementary file 1e*). Secondly, we calculated the heritability tagged by causal loci and PolyFun causal SNP in independent test set (defined as the $R^2$ of linear regression; Method), and found that causal loci tagged 38–251% more heritability than causal SNP (median = 151%; *Supplementary file 1e*). This was not an artifact of larger number of causal loci, since the Akaike information criterion (AIC) was similar between causal loci and causal SNP (paired *t*-test p = 0.36; *Supplementary file 1e*). Thirdly, for traits with sufficient causal loci coverage, we also applied Linkage Disequilibrium Score regression (LDSC) on independent GWAS summary statistic to evaluate heritability enrichment in causal loci. On average, causal loci showed 124-fold enrichment of heritability, significantly larger than genomic region-matched control loci (124- vs 101-fold; p = 0.0002, Method and *Figure 2—figure supplement 1*). Lastly, we applied simulation analysis and found that HFS + SUSIE showed similar advantages over SNP-based methods as in real data, with high accuracy and low false-positive rate (FDR) (Supplementary materials).

We further applied a sliding-window analysis (step = 2048 bp, Method) to test whether HFS-based result is robust against the choice of sequence interval. 29.4% of causal loci (PIP >0.95) in the original analysis were still causal in sliding-window analysis. 31.1% and 29.3% of causal loci whose 5′ and 3′ overlapping locus had PIP >0.95 in sliding-window analysis, respectively, while themselves were no longer causal. Besides, HFS + SUSIE was also robust when the predefined number of causal loci (L = 2–10) was changed, and the number of detected loci was not changed. Lastly, removing insertion and deletion would reveal 9% more significant association (p < $5 \times 10^{-8}$) but 4.7% less causal association (PIP >0.95), and slightly increased inflation factor (Wilcoxon p = 0.0001, *Figure 2—figure supplement 1*). Taken together, HFS-based SUSIE is a powerful and robust strategy for individual data-based genetic fine-mapping.

## Biological interpretation based on HFS

Pinpointing causal loci of complex traits provides the opportunity of analyzing the biological mechanism of them. Thus, based on the HFS-based fine-mapping result, we applied a linear regression model to analyze the underlying pathways, cell types, and tissues of each complex trait. For each locus, we annotated its relevance to a pathway by combined SNP to Gene (CS2G) strategy (*Gazal et al., 2022*), and regressed the PIP against this annotation, with a set of baseline annotations included as covariates, similar to the LDSC framework (*Finucane et al., 2018*) (Method). After p-value correction and recurrent pathway removal (Method), we detected a total of 727 pathway–trait associations (*Figure 3A* and *Supplementary file 1g*). The most significant associations were 'megakaryocyte differentiation' with platelet count (p = 2.26 × 10$^{-34}$), 'Insulin-like growth factor receptor signaling pathway', 'Endochondral ossification' with height (p = 4.95 × 10$^{-33}$ and 1.17 × 10$^{-27}$), 'PD-1 signaling' with allergic disease (p = 5.55 × 10$^{-25}$), and 'major histocompatibility complex pathway' with asthma (p = 1.22 × 10$^{-23}$). In fact, asthma and allergic disease were predominantly associated with more than 80 immune-related pathways. These associations were all in line with existing knowledge of trait mechanism, and extended the understanding of their genetic basis. For example, PD-1 has recently been suggested as potential targets of allergic diseases like atopic dermatitis (*Galván Morales et al., 2021*), but such association has not been highlighted by previous genetic association studies.

For other traits, the most significant associations also replicated known mechanisms, such as 'osteoblast differentiation', 'Wnt ligand biogenesis and trafficking' with BMD (p = 4.59 × 10$^{-13}$ and 2.78 × 10$^{-12}$); 'circadian pathway' with chronotype (p = 4.25 × 10$^{-12}$); 'calcium regulated exocytosis of neurotransmitter', 'Arachidonic acid metabolism' with intelligence (p = 5.52 × 10$^{-7}$ and 2.78 × 10$^{-6}$); 'GPCR pathway' and 'adipogenesis' with BMI (p = 4.97 × 10$^{-10}$ and 2.02 × 10$^{-7}$) and 'physiological cardiac muscle hypertrophy' with systolic blood pressure (p = 6.32 × 10$^{-11}$). We also highlighted less significant association which provided novel insights, such as 'synaptic vesicle docking' and 'neuron migration' with chronotype (p = 4.00 × 10$^{-7}$ and 4.55 × 10$^{-7}$), 'Prostaglandins synthesis' with insomnia (p = 5.30 × 10$^{-9}$), 'behavioral response to cocaine' with alcohol intake (p = 3.39 × 10$^{-8}$) and 'roof of mouth development' and 'glycoside metabolism' with forced vital capacity (FVC) (p = 2.19 × 10$^{-12}$ and 5.73 × 10$^{-11}$).

For cell type and tissue analysis (*Figure 3B* and *Supplementary file 1h*), we applied the same linear model to evaluate whether causal loci enriched in active chromatin regions of each cell type (Method). We found 153 biologically interpretable associations with complex traits. For example, fetal megakaryocyte (p = 5.67 × 10$^{-22}$) and child spleen (p = 2.15 × 10$^{-13}$) were found to be key cell type and tissue of platelet count. Systolic blood pressure was significantly associated with multiple heart and artery tissues and fetal cardiomyocyte (p < 1.63 × 10$^{-5}$), whereas allergic disease was associated with multiple immune cells including natural killer, Treg, and B cells (p < 4.79 × 10$^{-16}$). For brain-related traits, we found 21 significant associations, 14 of which were from central nervous system. For example, adult hippocampus and cingulate gyrus were both linked to alcohol intake, smoking, and insomnia (p < 1.11 × 10$^{-5}$), whereas chronotype was associated with embryonic brain germinal matrix (p < 8.68 × 10$^{-6}$) and intelligence with embryonic neuron-derived stem cell (p < 6.89 × 10$^{-7}$).

We also applied other modified strategies for this task but did not get satisfying result. For example, using cS2G to link locus to gene lists specifically expressed in each cell type suffered from scRNA dataset batch effect, whereas linear mix model was less sensitive than standard linear model (Supplementary Materials).

Taken together, our result suggests that fine-mapping results based on HFS could pinpoint the causal pathways, cell types, and tissues underlying complex traits, and is valuable for the biological interpretation of genetic association study.

## Highlighted genes for complex traits

Enhanced power of fine-mapping and biological enrichment could reveal novel key genes for trait mechanism study. Below we integrated fine-mapping result and their functional annotation in several case studies to find causal signals and trait-relevant genes in regions not resolved by previous genetic association studies.

In our study, platelet count had large number of causal loci (*Figure 2*) which showed significant functional enrichment (*Figure 3*). To find key loci and genes underlying platelet count, we focused on causal loci that overlapped with active regions in 'fetal megakaryocyte' and 'child spleen tissue', and

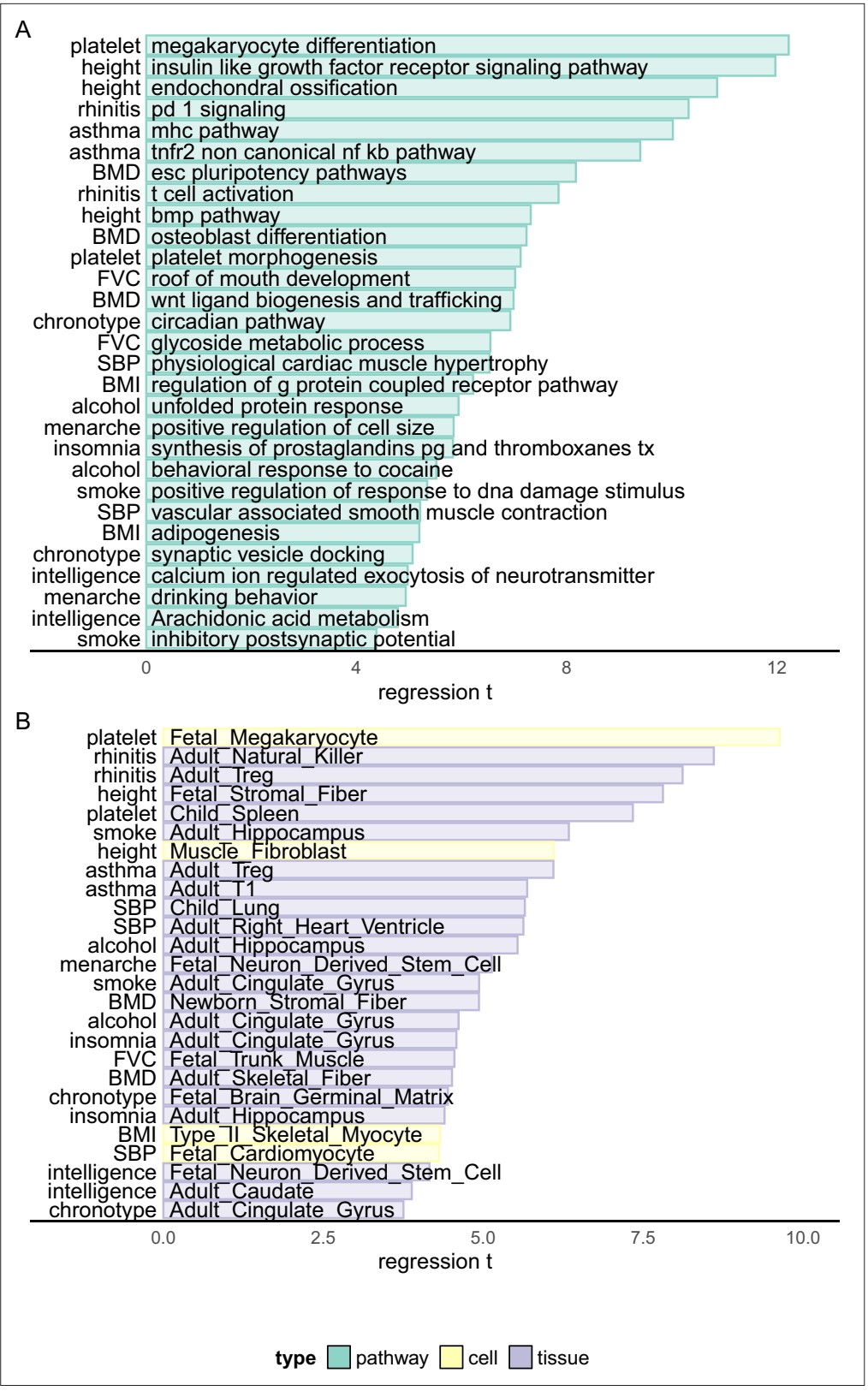

**Figure 3.** Biological enrichment analysis based on Haplotype Function Score (HFS) fine-mapping. *x*-axis indicated *t* statistics of the analyzed term in a multivariate linear regression (Method). Cell: single-cell ATAC peak for 222 cell types from *Zhang et al., 2021a*. Tissue: active chromatin regions of 222 tissues from epimap (*Boix et al., 2021*). For each trait, we showed the most significant term plus one or two terms with high biological interpretation that

*Figure 3 continued*

also passed significance threshold. Full enrichment result is shown in **Supplementary file 1g** and **Supplementary file 1h**.

applied cS2G (**Gazal et al., 2022**) to link them to two key pathways ('megakaryocyte differentiation' and 'platelet morphogenesis', Method and **Figure 4A**). We chose these annotations based on p-value in biological enrichment analysis in **Figure 3**. A total of 25 loci were highlighted (**Figure 4A**), which were recurrently linked to well-known platelet-regulating genes like MEF2C, SH2B3, FLI1, RUNX1, THPO, and NFE2. Among them we noticed a less-studied gene RBBP5, a target of key transcriptome factor MEF2C during megakaryopoiesis (**Kong et al., 2019**). Specifically, in 1q32.1 region, HFS + SUSIE identified two loci with PIP >0.9 (**Figure 4B**). SNP-based association also found significant association in this region, but SNP fine-mapping (**Weissbrod et al., 2020**) could not resolve this signal and only found seven signals between PIP = 0.1–0.5. This was unlikely a statistical inflation, since HFS-based association test p-value was actually higher than SNP-based one (**Figure 4—figure supplement 1**). One of the causal loci, chr1:47401806–47405902 (PIP = 1), overlapped with spleen active chromatin and harbored a cCRE in megakaryocyte, and was linked to RBBP5 and three other genes. RBBP5 is known to be involved in megakaryocyte differentiation during megakaryopoiesis and was regulated by MEF2C (**Kong et al., 2019**), but previous genetic association studies provided little evidence for its association with platelet count.

The major histocompatibility complex (MHC) region has long been a challenge of genetic association study due to its long-range LD, and is often excluded in fine-mapping tools. However, many disorders like schizophrenia (**Sekar et al., 2016**) and immune diseases (**Nawijn et al., 2011**) are robustly associated with MHC region. In our HFS-based fine-mapping of asthma, we found 15 loci within MHC region had PIP >0.95, 11 of which overlapped with active chromatin regions in Treg or natural killer cells (**Figure 4C** and **Supplementary file 1j**). This result showed good discrimination between causal and non-causal loci: despite these 15 likely causal loci, only six loci had PIP between 0.25 and 0.95. Since MHC region harbored a large number of genes, these causal loci were linked to as much as 105 potential target genes, which hindered the discovery of true targets. We further filtered them based on the involvement in pathway 'TNFR2-NFKB pathway' and 'innate lymphocyte [ILC] development', since these pathways were most significantly associated with asthma (**Figure 3**), even after excluding MHC region (p = $2.57 \times 10^{-13}$ and $1.39 \times 10^{-17}$). We found five genes (LTA, LTB, TNF, PSMB8, and PSMB9) that were predicted to be regulated by five causal loci overlapped with active chromatin regions (**Figure 4C**), which could be considered as potential key genes for further validation.

Similarly, we fine-mapped MHC region for other allergic diseases (**Figure 4—figure supplement 2** and **Supplementary file 1j**) and found potential key genes including HLA family and AGER. We also highlighted other gene–trait association not previously emphasized by GWAS, including GATA4 and NPPA (cardiac muscle hypertrophy) with SBP, ALOX5 (arachidonic acid metabolism) with intelligence and CRY1 (circadian pathway) with chronotype, as further discussed in **Supplementary file 1k, l, m** and supplementary information.

On the other hand, HFS perform worse than SNP-based fine-mapping on exonic regions. Taking height as an example, PolyFun detected 125 causal SNPs (PIP >0.95) in the exonic regions, but only 16% (20) of loci that harbored them also reached PIP >0. 5 (11 reached PIP >0.95) in HFS + SUSIE analysis. Among the 105 loci that missed such signals (HFS PIP <0.5), 12 had a nearby loci (within 10 kb) showing HFS PIP >0.95, which likely reflected false positive led by LD. Thus, SNP-based analysis should be prioritized over HFS in coding regions.

## HFS-based polygenic prediction

Lastly, we analyzed the potentiality of HFS in polygenic prediction accuracy. Compared with state-of-the-art SNP-based polygenic risk score (PRS) algorithm LDAK-BOLT (**Zhang et al., 2021b**), HFS-based PRS (weighted by SUSIE posterior effect size) reached 47–90% of $R^2$ in independent European test set (meta-analyzed proportion = 75.6%, 95% confidence interval = 75.3–75.8%, **Figure 5—figure supplement 1**). The gap between performance of HFS- and SNP-based PRS reflected the fact that HFS only captured (the majority of) functional genomic alterations and missed the information of amino acid sequence and post-translational modification. We thus proposed that integrating information from HFS and SNP could provide better performance. Specifically, in the large European training set we

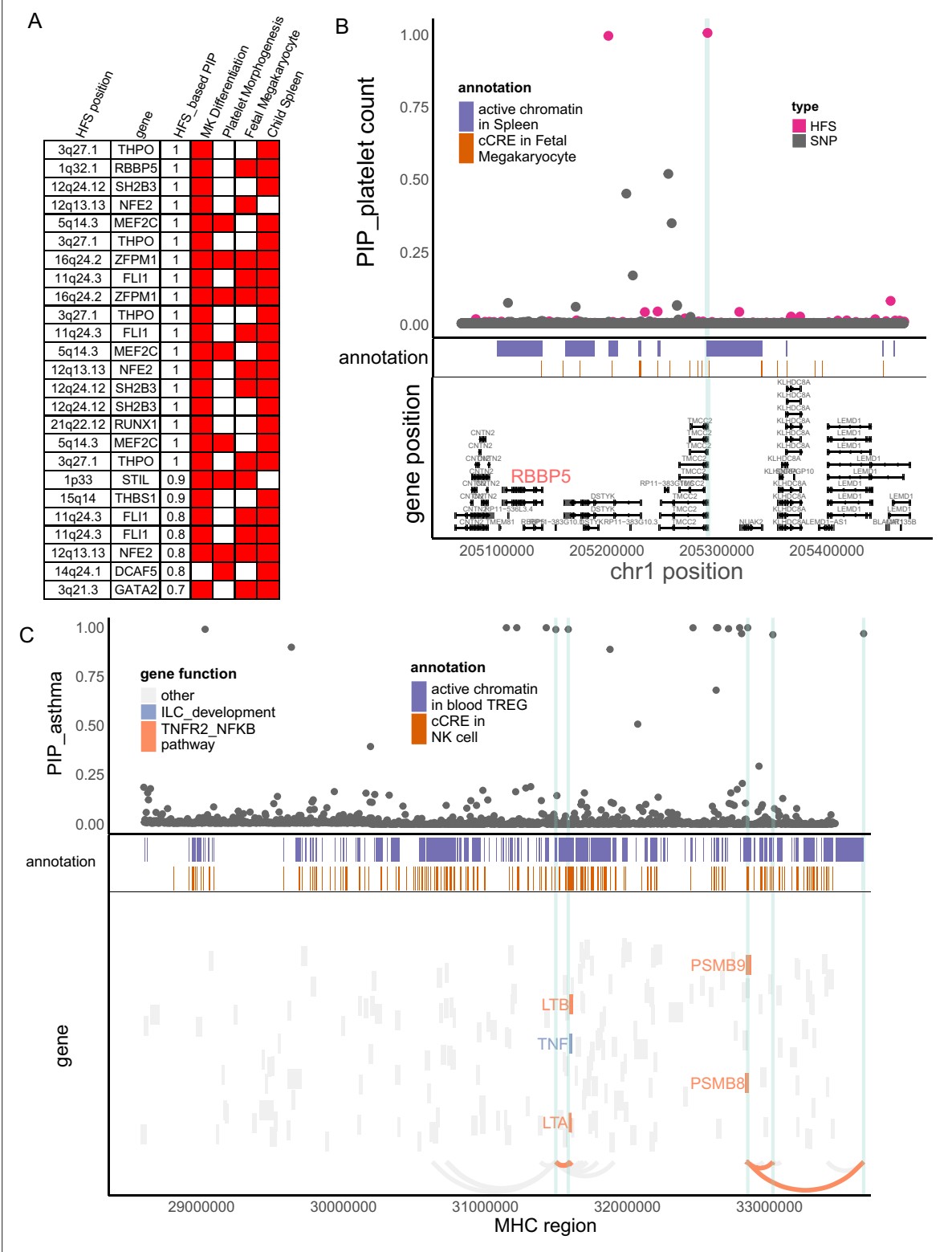

**Figure 4.** Haplotype Function Score (HFS) linked trait to causal genes. (**A**) Target genes of causal loci identified by HFS + SUSIE for platelet count. Only genes that showed functional convergence were shown. (**B**) Regional plot for RBBP5. HFS: loci posterior inclusion probability (PIP) calculated by HFS + SUSIE. SNP: SNP PIP calculated by PolyFun. cCRE: credible cis-regulation elements. (**C**) Regional plot of major histocompatibility complex (MHC) region for asthma. Thickened curve linked highlighted causal loci to its target genes predicted by cS2G (*Gazal et al., 2022*).

The online version of this article includes the following figure supplement(s) for figure 4:

*Figure 4 continued on next page*

*Figure 4 continued*

**Figure supplement 1.** p-value comparison within RBBP5 region between Haplotype Function Score (HFS) and SNP association test.

**Figure supplement 2.** Locus analysis for allergic diseases.

trained SNP PRS model by LDAK. Then, in a small tuning sample of target ancestry, we calculated per-locus HFS prediction score of height (sum of HFS within this block, weighted by SUSIE posterior effect size), then used machine learning to integrate them with LDAK PRS into a final polygenic prediction score, hereafter referred to as 'HFS + LDAK'. To choose the proper machine-learning tools to achieve this goal, in British European test set we applied LASSO, ridge regression, and elastic net and compared the result (*Figure 5B*). They gave comparable result with only difference of $R^2$ around

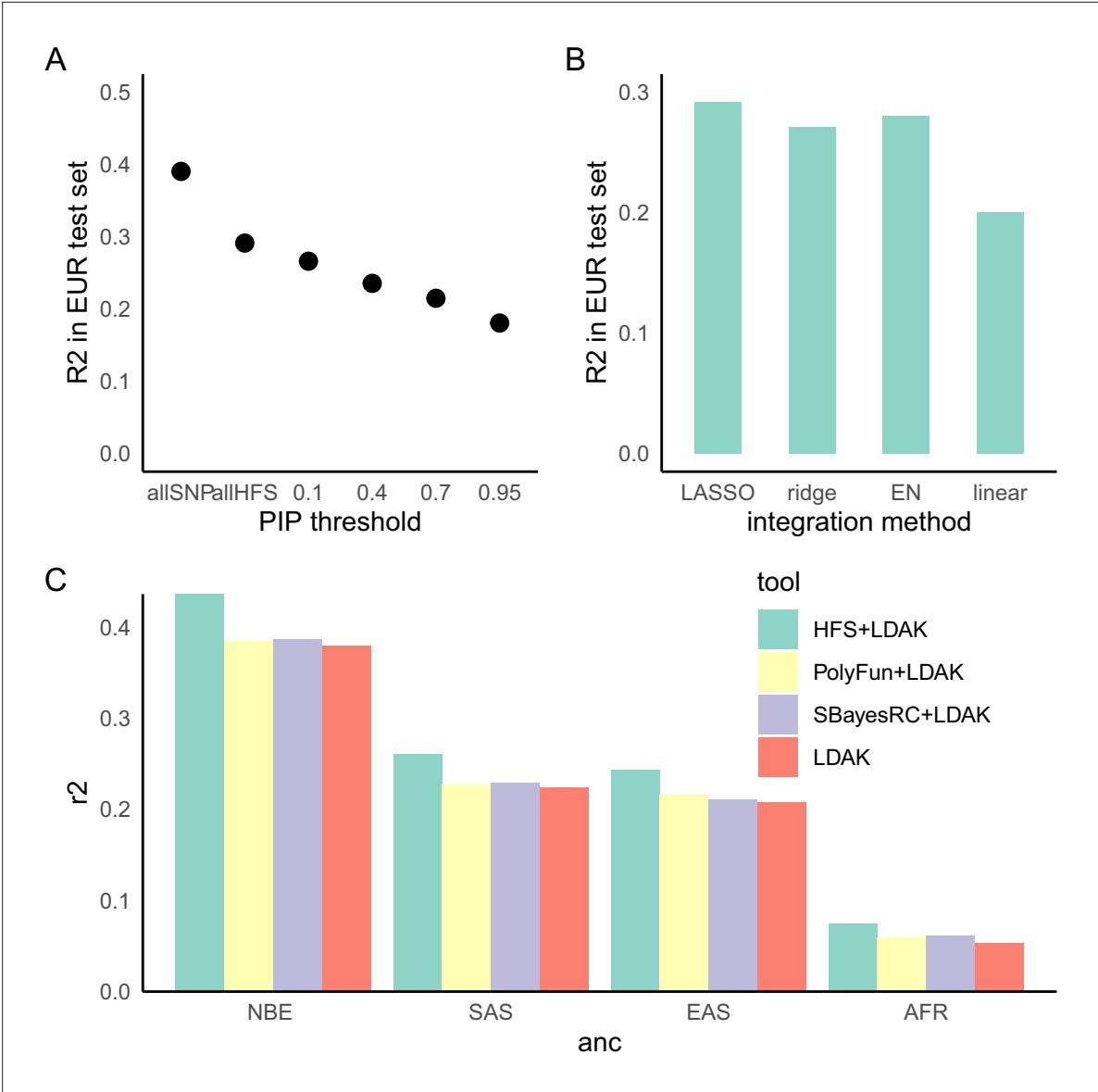

**Figure 5.** Haplotype Function Score (HFS)-based polygenic prediction. (**A**) Prediction $R^2$ of HFS-based polygenic risk score (PRS) using different threshold of posterior inclusion probability (PIP). allSNP: SNP-based PRS calculated by LDAK-BOLT (*Zhang et al., 2021b*). *n*: number of features included in the corresponding PRS. (**B**) Prediction $R^2$ of per-block HFS score in British European test set by different methods. EN: elastic net. (**C**) Prediction $R^2$ of different tools in non-British European (NBE), South Asian (SAS), East Asian (EAS), and African (AFR) groups in UK Biobank.

The online version of this article includes the following figure supplement(s) for figure 5:

**Figure supplement 1.** Proportion of heritability captured by Haplotype Function Score (HFS) polygenic risk score (PRS).

0.01, and all of them were profoundly better than simple linear regression. We chose LASSO as the algorithm in the formal analysis.

Using height as a representative trait, we first estimated the proportion of variance captured by top loci, and found that HFS of loci with PIP >0.4 ($n$ = 5101) captured roughly 80% of variance explained by all genome-wide loci ($n$ = 1,200,024 corresponded to sling-window strategy; *Figure 5A*). We then calculated HFS + LDAK in non-British European (NBE), South Asian (SAS), East Asian (EAS), and African (AFR) population in UK Biobank, and observed 17.5%, 16.1%, 17.2%, and 39.8% improvement over LDAK alone (p = 3.21 × $10^{-16}$, 0.0001, 0.002, and 0.001, respectively. *Figure 5C*). As a comparison, we integrated LDAK with PolyFun-pred (*Weissbrod et al., 2022*) and SbayesRC (*Zheng et al., 2022*) using Polypred framework (*Weissbrod et al., 2022*), but did not observe significant improvement over LDAK alone (difference in $R^2$ < 0.01, p = 0.001–0.07, *Figure 5C*). Since PolyFun-pred + BOLT-LMM has been shown to significantly outperformed BOLT-LMM alone (*Weissbrod et al., 2022*), we reasoned that the improvement of LDAK over BOLT-LMM might have attenuated the improvement brought about by PolyFun-pred, making it difficult to reach significance threshold. Taken together, we concluded that HFS could bring about mild but significant improvement to classic SNP-based PRS in the task of cross-ancestry polygenic prediction.

## Discussion

In this study, we designed the new HFS framework for genetic association analysis and demonstrated that it could improve classic SNP-based analysis in terms of causal loci and gene identification, biological interpretation and polygenic prediction. We suggest that HFS is a promising strategy for future genetic studies, but more progresses in algorithm and computation and data resources are still desired.

Compared with SNP, HFS has several compelling features. For instance, LD between adjacent HFS is much lower than SNP, which enhances the precision of statistical fine-mapping. For those false-positive variants caused by LD, they are expected to make little impacts on functional genomics, thus their HFS would be close to reference and would not influence downstream analysis significantly. In line with these advantages, we showed that HFS-based fine-mapping had high statistical power, and downstream enrichment analysis was capable of revealing biologically interpretable mechanisms. As a typical example, our findings of enrichment of intelligence-associated loci in arachidonic acid metabolism pathway is in line with the well-known role of polyunsaturated fatty acid in neurodevelopment (*Helland et al., 2003*). Nonetheless, previous GWAS provided little evidence on this association. Secondly, HFS could integrate effects of all variants within a locus, regardless of their population frequency. Thus, HFS could capture information from rare variants overlooked by classic association study and improve polygenic prediction, as shown by our result. In fact, HFS framework could directly extend to whole-genome sequencing data and capture all mutations as rare as singleton, making one step forward to fill in the 'missing heritability'.

Despite its potential, the current HFS framework carries several drawbacks and necessitates significant enhancements. A key limitation is the substantial computational cost. In this study, the transformation phase of the genotype–haplotype sequence for UK Biobank SNP data required hundreds of thousands of CPU core hours. This computation cost would increase exponentially when analyzing whole-genome sequencing data or employing a sliding-window strategy. A potential solution could involve developing a new algorithm that bypasses the variant calling stage and directly generates DNA sequences per locus from raw sequencing or SNP array data. For the sequence-to-HFS step, Sei (*Chen et al., 2022*) required about 1.8 GPU hours per one million sequences. Intriguingly, the majority of Sei's output is unused in the HFS framework, since Sei predicts over 20,000 functional genomic features, while the HFS only represents one of their integrated scores. Future development of novel DL models that predict functional genomics in a manner more fitting to the HFS framework could considerably reduce computation costs. Lastly, it is currently unfeasible to incorporate all genome-wide HFS into a single LASSO model. This limitation forced us to first integrate HFS into pre-locus score, which inevitably sacrificed the accuracy.

Another hurdle arises in integrating HFS with other genomic features. Intrinsically, HFS captures only the variant effect mediated by functional genomics, while a genetic variant might also influence amino acids, post-transcriptional modifications (PTMs) (*Park et al., 2021*), and 3D chromosomal structures (*Zhou, 2022*). Therefore, HFS alone cannot wholly replace SNP without any loss, as our results

demonstrate that the HFS-based prediction model captured approximately 70% of the variance explainable by the SNP-based prediction model. One potential solution is to extend the concept of HFS, applying DL to quantify the genetically determined values of PTMs, protein biochemical properties (*Pejaver et al., 2020*), and protein and chromosomal structures, potentially employing AlphaFold (*Jumper et al., 2021*)-derived features (*Liu et al., 2022*). Analyzing HFS in conjunction with these multi-modal function scores could provide a comprehensive depiction of the genetic architecture of complex traits. However, the colossal computational cost is currently prohibitive. As a compromise, we simply performed joint analysis of HFS with SNP PRS in our prediction model analysis. This approach is far from optimal, as it led to only moderate improvement and did not enhance fine-mapping and biological enrichment analysis.

The challenge of using sequence-based DL models in HFS applications is further compounded by their difficulty in predicting variations between individuals. Recent studies (*Huang et al., 2023*; *Sasse et al., 2023*) indicate that DL models, trained on the reference human genome, demonstrate limited accuracy in predicting gene expression levels across different individuals. This limitation is likely due to the models' inability to account for long-range regulatory patterns, which are crucial for understanding the impact of variants on gene expression and vary across genes. In contrast, our study leveraged sequence-determined functional genomic profiles in association studies, which mitigates this issue to an extent. For instance, although sei cannot identify the specific gene regulated by a given input sequence, it can predict changes in the sequence's functional activity. Future improvements in DL models' ability to predict interindividual differences could be achieved by incorporating cross-individual data in the training process. An example of such data is the EN-TEX (*Rozowsky et al., 2023*) dataset, which aligns functional genomic peaks with the specific individuals and haplotypes they correspond to.

In summary, our results demonstrate that incorporating HFS to represent genetically determined functional genomic activities in genetic association studies offers robust improvements in both the biological interpretation and polygenic prediction of complex traits. Thus, the application of the HFS framework in future genetic association studies holds considerable promise.

## Methods

### Sample description

This study analyzed UK Biobank data, with application ID 84436, and was adhered to the ethics and privacy policy of UK Biobank. We only included participants with array imputed genotype data in bgen format that passed UKB quality control, and removed related individuals. We randomly selected 350,587 self-identified British ancestry Caucasians as training sample. The remaining participants were grouped according to their ancestry, where non-British European, South Asian, East Asian, and African groups serve as test samples.

All phenotypes analyzed (*Supplementary file 1b*) were collected from UKB table browser, which came from self-report or physical measurement. Phenotypes were first adjusted by age, sex, top 10 principal components, Townsend index, and genotype array quality metrics by linear regression. We then applied inverse-normal transformation on the residuals. Binary phenotypes were adjusted in the same way except by generalized linear regression.

### Genotype data processing

We first segmented hg38 genome into 4096 bp loci. To do so, we downloaded chromatin state annotation of 222 human tissues at different developmental stage (embryo, newborn, and adult) from epimap (*Boix et al., 2021*) database. For each tissue, all chromosomal regions annotated as 'transcription start site (TSS), transcription region (TX), enhancer, promoter' in at least half of the samples were marked as active regions. The union of active regions across all tissues was taken, and regions annotated as genomic gaps (centromere, ambiguous base pairs, etc.) in the Hg38 genome were removed. Then, for this series of active regions, if the length is less than 4096 bp, the locus is defined as a 4096-bp area centered around the active region. If the length is greater than 4096 bp, 4096-bp length loci are gradually delineated from the midpoint outward. Finally, non-overlapping 4096 bp blocks were used to cover the remaining genomic regions. This resulted in about 617,378 genomic regions in total. In the sliding-window analysis, all these blocks were shifted 2048 bp toward 5' end,

generating another 617,378 blocks. We repeated the fine-mapping analysis and applied polygenic analysis on these combined blocks, using height as a representative trait.

For each of the loci, we obtained ID of variants within this locus by bedtools (**Quinlan and Hall, 2010**), then extracted genotypes from UKB.bgen file by bgenix, finally used Plink (**Purcell et al., 2007**) to remove all variants with INFO <0.8, Hardy–Weinberg p < $10^{-6}$, allele count <10 or missing rate >10%, and removed individual that missed more than 10% of retained variants in this locus. The output vcf file was liftover to hg38 by Crossmap (**Zhao et al., 2014**) and phased by SHAPEIT4 (**Delaneau et al., 2019**). Phased vcf was transformed to.haps format by Plink, which in turn gave rise to two files: a vcf file containing information of each haplotype, and an n x 2 matrix in plain text that recorded the id of two haplotypes per individual.

## HFS calculation

There has been several DL models that predict functional genomic profiles based on DNA sequence (**Avsec et al., 2021**; **Chen et al., 2022**; **Kelley, 2020**; **Yan et al., 2021**; **Zhou et al., 2018**). Among them, we chose sei (**Chen et al., 2022**) to calculate HFS for the following reasons: (1) the required input length (4096 bp) is moderate; (2) it represents 21,906 functional genomic tracks, more comprehensive than other models; (3) it integrated information of the entire sequence, not only the few bp at the center. For each haplotype at each locus, we generated its corresponding DNA sequence by bcftools (**Danecek et al., 2021**) consensus option. At each locus, the start point of each sequence was matched to the start point of reference sequence. When insertion variants made the sequence longer than 4096 bp, we discarded base pairs at the 3′ end. Likewise, with deletion variants, we added N to the 3′ end. We applied sei to predict 21,906 functional genomic tracks for each sequence, without normalizing for histone mark (divided each track score by the sum of histone mark score) as suggested by the sei author. We then used the projection matrix provided by sei to calculate forty sequence class scores, which could be regards as the weighted sum of these tracks and represented different aspect of functional genomic activities. We discarded the last score (heterochromatin 6 [centromere]), since its proportion is too low and is functionally trivial, leading to 39 scores per haplotype.

On each individual, we derived from each sequence class score the mean of two haplotypes, corresponding to additive model. For HFS LD calculation, we extracted the mean value of sequence class score corresponding to reference sequence class of adjacent loci, and calculate $R^2$ value between them. The sequence class score of the reference sequence class was defined as the HFS for this locus, and was used for downstream trait association analysis.

## HFS–trait association

For each locus, we calculated the association between trait-specific HFS and adjusted, normalized trait value by linear regression, without any covariates (this is because all selected covariates have been adjusted at the normalization step). For uniformity, we set the significance threshold at p < 5 × $10^{-8}$, even if it was over-stringent for $n$ = 590,959 loci. Among significant associations, we defined an independent association as the locus with the lowest p-value in the 200 kb regions. As a positive control, we applied quantitative and binary GWAS with REGENIE (**Mbatchou et al., 2021**), using default settings and the same British training sample. The main difference is that we used raw trait values in REGENIE, and provided the same covariates. We calculated the genomic control inflation factor, $\lambda_{GC}$, as the median of $X^2$ statistics, separately for HFS association test and GWAS only those SNPs in hapmap3 (**Altshuler et al., 2010**) project were calculated. We compared the $\lambda_{GC}$ between HFS and SNP by Pearson correlation analysis and paired $t$-test.

## Fine-mapping analysis

We divided hg38 genome into 1361 independent blocks as defined by **MacDonald et al., 2022**, and applied SUSIE to HFS of all loci within each block, separately for each trait (parameters: maximum number of causal signal = 10, coverage = 0.95). We subtracted reference HFS value for each locus prior to analysis, such that homozygous reference haplotype corresponded to HFS = 0. To avoid influence of sei prediction noise, we rounded the HFS value at two decimals. This is due to the fact that even if a variant actually makes no impact on functional genomics, Sei would still output a value that are close to but not equal to reference sequence class score. Rounding procedure would set such HFS to zero and remove the random value from sei. Loci whose HFS had PIP >0.95 were defined as

causal loci, and loci that had causal association with multiple traits were defined as pleiotropic loci. As a positive control, we applied PolyFun (*Weissbrod et al., 2020*) and SbayesRC (*Zheng et al., 2022*) on the GWAS summary statistics by REGENIE on the same training set, and extracted the reported PIP to define causal SNP.

To analyze the functional characteristics of causal loci, we first defined the sequence class of each locus by the maximum sequence class score of reference haplotype. We then tested whether each sequence class contained excess causal loci of each trait by Fisher's test. For each causal locus, we also defined a 'control' locus as the nearest locus that matched the p-value of this causal locus, and tested whether causal loci carried more PolyFun causal SNP than control loci by Fisher's test. Furthermore, For traits whose causal loci covered >0.1% of genome-wide SNP, we applied LDSC (*Finucane et al., 2015*) to quantify the heritability enrichment in causal and control loci, and compare their difference by jackknife method. To avoid winner's curse, we used external GWAS summary statistics for this analysis (*Mikaelsdottir et al., 2021*; *Yengo et al., 2022*). As an alternative method to quantify the heritability captured by causal loci, we ran multivariate linear regression in independent British test set where HFS of causal loci were independent variables and trait value were dependent variable, and calculated the $R^2$ and AIC. We applied the same analysis on causal SNP, and compared AIC between HFS and SNP multivariate regression.

## Functional enrichment analysis

Similar to the idea of LDSC (*Finucane et al., 2015*), we first generated a series of baseline annotation of each locus, then tested whether locus PIP was associated with functional annotations after controlling the impact of these baseline annotations. Specifically, we defined the following baseline annotations:

1. Number of haplotypes, range of HFS distribution of all haplotypes (scaled by reference HFS), and 39 sequence class score of reference haplotype.
2. Genomic regions of conserved base, high Phastcons score (*Siepel et al., 2005*) in mammals, primates and vertebrate, exon, intron, untranslated regions at 3′ and 5′ and 200 bp flanking regions of TSS. We used bedtools *intersect -f 0.1* option to annotate each locus by these annotations.
3. Maximum *B* statistics (*McVicker et al., 2009*), minimum allele age, and ASMC$_{avg}$ (*Palamara et al., 2018*) of all variants within this locus.

Type 2 and 3 annotations were directly obtained from LDSC (*Finucane et al., 2015*) baseline annotations. We did not include annotations related to functional genomics, since 39 sequence class scores were used to capture functional genomic characteristics. Conditioned on these baseline annotations, we analyzed the enrichment of PIP in the following functional annotations:

1. Biological pathways: We downloaded all pathways from MsigDB (*Subramanian et al., 2005*), C2: canonical pathways category (including Reactome (*Fabregat et al., 2018*), Pathway Interaction Database (PID) (*Schaefer et al., 2009*), Biocarta and Wikipathway) and C6: Gene ontology (*Ashburner et al., 2000*) (biological process) category. We retained only pathways with >5 and <500 genes. We generated a gene × pathway binary matrix and applied hierarchical clustering so that similar pathways were placed close to each other. We sequentially compared adjacent pathways, and removed the smaller one if the fraction of overlap >30%. A total of 3219 pathways were retained. We then linked each locus to these pathways by cS2G (*Gazal et al., 2022*) strategy. Specifically, a locus L would be annotated as 1 for pathway P only if L contained a SNP that was link to P with cS2G score >0.5.
2. Tissue-specific chromatin activity: We downloaded chromHMM (*Ernst and Kellis, 2012*) chromatin state annotation for 833 samples from epimap (*Boix et al., 2021*), and grouped them according to developmental stages and second-level tissue types. For each group, all chromosomal regions annotated as 'transcription start site (TSS), transcription region (TX), enhancer, promoter' in at least half of the samples were marked as active regions. We used bedtools *intersect -f 0.1* option to annotate whether each locus was active in each tissue.
3. Cell type-specific open chromatin regions: We downloaded scATAC-seq peak data from *Zhang et al., 2021a*, and annotated each locus by bedtools *intersect -f 0.1* option.

We applied multivariate linear regression of PIP against baseline annotations +one of the functional annotations. Regression coefficient >0 and Bonferroni-adjusted regression p-value <0.05 were used as significance threshold. From the final results, we manually removed those pathways and cell types

that reached significance threshold in more than half of the traits, since these pathways likely reflected unrecognized confounders.

## Polygenic prediction

We used the posterior effect size estimated by SUSIE on sliding-window strategy (doubling the number of loci) as weights, and calculated the weighted sum of HFS as the PRS of each trait, and calculated $R^2$ in independent British test sample with simple linear regression. As a positive control, we applied LDAK-BOLT (*Zhang et al., 2021a*) algorithm on the SNP array data (about seven hundred thousand variants) with tenfold cross-validation and max iteration = 200 in the same training sample, and calculated SNP-based PRS with the output SNP weights. Normalized trait values were analyzed, without any covariates provided. Array data were filtered by Plink with option `--geno` 0.1 `--hwe` 1e-6 `--mac` 100 `--maf` 0.01 `--mind` 0.1.

To train the refined model that predict height, we first calculated per-block HFS-based prediction score of height as the weighted sum of HFS within this block. Then, within each target ancestry group (non-British European (NBE), South Asian (SAS), East Asian (EA), and African (AFR) participants in UK Biobank), we randomly selected half as tuning sample and half as test sample. In the tuning sample, we applied LASSO regression that included both LDAK PRS and genome-wide per-block HFS score (1361 in total). The choice of LASSO regression was based on a comparison on British European test set (*Figure 5B*), where LASSO, ridge, and elastic net gave similar results and LASSO was relatively better. In the tuning sample of target ancestry, LASSO estimated the weights to combine per-block HFS score and LDAK PRS. We calculated the final prediction score in the test sample using these weights, and evaluated its prediction by linear regression $R^2$. Since the outcome (height) has already been adjusted and standardized, no covariates were included in this step. Additionally, we applied PolyFun-pred (*Weissbrod et al., 2022*) and SbayesRC (*Zheng et al., 2022*) to the summary statistics of height (calculated by REGENIE in the same training sample), and integrated their effect size with LDAK weight in the tuning sample using Polypred (*Weissbrod et al., 2022*) method. PRS for LDAK, LDAK + PolyFun and LDAK + SbayesRC were calculated by plink *score* option, excluding variants with INFO <0.8, Hardy–Weinberg $p < 10^{-6}$, allele count <2 or missing rate >10% in the target test set.

## Simulation analysis

We simulated trait levels using HFS data from chromosome 1 in a randomly selected 50,000 samples from UKB EUR training data. We randomly selected 1% (500) loci, assigned effect size from standard normal distribution, and calculated the aggregated genetic liability. We then simulated trait levels with $h^2 = 0.1$. We applied HFS + SUSIE as well as REGENIE + PolyFun on simulated traits and calculated the area under curve (AUC), FDR at PIP >0.95 for HFS + SUSIE. We repeated this procedure for 30 times.

On average, HFS + SUSIE showed high accuracy in identifying causal loci (median AUC = 0.92) and the FDR at PIP >0.95 is median 0.059. In line with real data analysis, the number of causal loci identified by HFS + SUSIE is 1.12-fold more than PolyFun on average. Furthermore, HFS + SUSIE showed good discrimination between causal and non-causal loci: the number of PIP >0.95 loci is larger than 0.5 < PIP < 0.95 loci.

## Alternative strategy on biological enrichment analysis

Despite the standard linear regression as we applied in the main text, we also applied a linear mixed regression which took independent blocks as random effect. For each regression, we included one biological term plus all baseline annotations. The regression coefficient and p-value of each biological term were estimated by mgcv R package. After p-value correction, most of the significant terms were those recurrently appeared in more than half of the traits, which were considered artifacts of hidden covariates. When removing these recurrent terms, less than five significant terms remained for each trait. We concluded that linear mixed regression was less sensitive than standard linear regression for identifying trait-specific biological association.

We also tried another strategy for cell type-specific analysis. We first downloaded C8 category from MsigDB, which contained gene lists specifically expressed in about 800 cell types, derived from multiple single-cell RNA sequencing studies. We then linked each locus to these gene lists by CS2G method, then applied linear regression, similar to pathway analysis. We found that most traits predominantly linked to nearly all cell types from a specific study, which showed study

batch effect instead of biological functions. For example, smoking was associated with all neuron subtypes, pericytes and immune cells from one brain scRNA data, but did not showed association with immune cells and pericytes from other scRNA studies. We reasoned that the curated cell type-specific gene lists contained batch effects that were not yet corrected. Thus, in the main text, we reported association between PIP and single-cell ATAC peak from one study, which reduced the batch effect.

## Highlighted genes for complex traits

For chronotype, we found one circadian gene CRY1 that were predicted to be target of locus chr12:1070930221107097118, which had PIP = 0.56. This locus was active in cingulate gyrus, and belong to sequence class 'enhancer-multi tissue'. CRY1 was known to participate in circadian pathway, and was not highlighted by previous GWAS. SNP-based fine-mapping also found no SNP with PIP >0.1 that was predicted to link to CRY1. We suggested that it was a novel promising target gene for understanding mechanisms of chronotype.

For systolic blood pressure, we found chr8:11726583–11730679 (PIP = 0.999) that resided on gene GATA4. This locus was active in both adult heart ventricle and in fetal cardiomyocyte. GATA4 took part in physiological myocardial hypertrophy. SNP fine-mapping got PIP <0.34 for all SNPs linked to GATA4. Previous GWAS has found its homolog GATA2 as a key gene in blood pressure, and our new result supported GATA4 as another key genes.

For intelligence, we found chr10:45559452–45563548 that was active in caudate nucleus and was associated with intelligence at PIP >0.5. It was predicted to regulate ALOX5, a key enzyme in the arachidonic acid metabolism. It is known that supplement of Arachidonic acid is beneficial for child intelligence development, and that arachidonic acid takes part in neurodevelopment. However, few genes related to arachidonic acid has been associated with intelligence.

## Statistical analysis

All p-values were two-sided and adjusted by Bonferroni unless otherwise specified. For group comparison, we used Fisher's test for count data and paired $t$-test for continuous data. For $R^2$ of PRS comparison, we applied r2redux (*Momin et al., 2023*) R package to estimate 95% confidence interval and its p-value for the difference of $R^2$.

## Acknowledgements

This work was supported by grants from the 2030 Science and Technology Innovation Key Program of Ministry of Science and Technology of China (No. 2022ZD020910001), the National Natural Science Foundation of China (No. 81971292, 82150610506) and the Natural Science Foundation of Shanghai (No. 21ZR1428600), the Medical-Engineering Cross Foundation of Shanghai Jiao Tong University (No. YG2022ZD026).

---

## Additional information

### Funding

| Funder | Grant reference number | Author |
|---|---|---|
| Ministry of Science and Technology | 2030 Science and Technology Innovation Key Program 2022ZD020910001 | Guan Ning Lin |
| National Natural Science Foundation of China | 81971292 | Guan Ning Lin |
| National Natural Science Foundation of China | 82150610506 | Guan Ning Lin |
| Natural Science Foundation of Shanghai | 21ZR1428600 | Guan Ning Lin |

---

| Funder | Grant reference number | Author |
|---|---|---|
| Medical-Engineering Cross Foundation of Shanghai Jiao Tong University | YG2022ZD026 | Guan Ning Lin |

The funders had no role in study design, data collection, and interpretation, or the decision to submit the work for publication.

## Author contributions

Weichen Song, Conceptualization, Data curation, Software, Formal analysis, Investigation, Visualization, Methodology, Writing – original draft, Project administration, Writing – review and editing; Yongyong Shi, Conceptualization, Resources, Supervision, Investigation, Writing – review and editing; Guan Ning Lin, Conceptualization, Supervision, Funding acquisition, Validation, Investigation, Project administration, Writing – review and editing

## Author ORCIDs

Weichen Song https://orcid.org/0000-0003-3197-6236

Reviewer #1 (Public review): https://doi.org/10.7554/eLife.92574.3.sa1
Reviewer #2 (Public review): https://doi.org/10.7554/eLife.92574.3.sa2
Author response https://doi.org/10.7554/eLife.92574.3.sa3

# Additional files

## Supplementary files
- MDAR checklist
- Supplementary file 1. Supplementary tables.

## Data availability

The current manuscript is a computational study, so no data have been generated for this manuscript. Data from the UK Biobank (project 84436; *Bycroft et al., 2018*) are available, pending application approval from: https://www.ukbiobank.ac.uk/. Modeling code is available at https://github.com/WeiCSong/HFS (copy archived at *Song, 2024*).

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
