## [Editor Report · eLife assessment]

This **valuable** paper presents a new approach for association testing, using the output of neural networks that have been trained to predict functional changes from DNA sequences. As such, the approach is an interesting addition to statistical genetics, and the evidence for the presented method being able to identify trait-associations in regions where GWASs are typically underpowered is **solid**. A limitation is, however, that it is unclear how the quality of these associations compares to those detected using conventional methods. Additional work assessing this method's power and characterizing false positives / false negative regions would be critical to ensure that the method is broadly adopted by the field.

---

## [Referee Report · Reviewer #1 (Public review)]

Summary:

In this paper, Song, Shi, and Lin use an existing deep learning-based sequence model to derive a score for each haplotype within a genomic region, and then perform association tests between these scores and phenotypes of interest. The authors then perform some downstream analyses (fine-mapping, various enrichment analyses, building polygenic scores) to ensure that these associations are meaningful. The authors find that their approach allows them to find additional associations, the associations have biologically interpretable enrichments in terms of tissues and pathways, and can slightly improve polygenic scores when combined with standard SNP-based PRS.

Strengths:

- I found the central idea of the paper to be conceptually straightforward and an appealing way to use the power of sequence models in an association testing framework.

- The findings are largely biologically interpretable, and it seems like this could be a promising approach to boost power for some downstream applications.

Weaknesses:

- While not a weakness of the manuscript, the proposed method is computationally intensive.

---

## [Referee Report · Reviewer #2 (Public review)]

Summary:

In this work, Song et al. propose a locus-based framework for performing GWAS and related downstream analyses including finemapping and polygenic risk score (PRS) estimation. GWAS are not sufficiently powered to detect phenotype associations with low-frequency variants. To overcome this limitation, the manuscript proposes a method to aggregate variant impacts on chromatin and transcription across a 4096 base pair (bp) loci in the form of a haplotype function score (HFS). At each locus, an association is computed between the HFS and trait. Computing associations at the level of imputed functional genomic scores enables integration of information across variants spanning the allele frequency spectrum and bolster the power of GWAS.

The HFS for each locus is derived from a sequence-based predictive model - Sei. Sei predicts 21,907 chromatin and TF binding tracks, which can be projected onto 40 pre-defined sequence classes ( representing promoters, enhancers etc.). For each 4096 bp haplotype in their UKB cohort, the proposed method uses the Sei sequence class scores to derive the haplotype function score (HFS). The authors apply their method to 14 polygenic traits, identifying ~16,500 HFS-trait associations. They finemap these trait-associated loci with SuSie, as well perform target gene/pathway discovery and PRS estimation.

Strengths:

Sequence-based deep learning predictors of chromatin status and TF binding have become increasingly accurate over the past few years. Imputing aggregated variant impact using Sei, and then performing an HFS-trait association is therefore an interesting approach to bolster power in GWAS discovery. The manuscript demonstrates that region-level associations can be identified at the level of an aggregated functional score using sequence-based deep learning models. The finemapping and pathway identification analyses suggest that HFS-based associations identify relevant causal pathways and genes from an association study. Identifying associations at the level of functional genomics increases portability of PRSs across populations. Imputing functional genomic predictions using a sequence-based deep learning model does not suffer from the limitation of TWAS where gene expression is imputed from a limited size reference panel such as GTEx and is an interesting direction to bolster discovery power.

However, a few limitations to this method in its current form are:

(1) HFS-based association is going to miss coding variation as well as noncoding regulatory variants such as splicing variants/polyadenylation variants which are not modeled by Sei. This will lead to false negatives in the HFS-based association and additionally false negatives + associated false positives in the finemapping. Going forward, it'll therefore be important to characterize how this influences the genome-wide finemapping.

(2) Sei predicts chromatin status / ChIP-seq peaks in the center of a 4kb region. It is thus not clear therefore whether the functional effects of variants not in the center of the 4kb region would be captured in a single Sei score. It also remains unclear how much the choice of window affects the association tests / finemapping.

(3) There are going to be cases where there's an association driven by a variant that is correlated with a Sei prediction in a neighboring window. These would represent false positives for the method, it would be useful to identify or characterize these cases.

Minor Concerns:

(1) Sequence based deep learning model predictions can be miscalibrated for insertions and deletions (INDELs) as compared to SNPs. It'll be important to note that model INDEL scores may not be calibrated, which might also lead to false positives / false negatives in the finemapping.

---

## [Author Response]

The following is the authors’ response to the original reviews.

**Public Reviews:**

**Reviewer #1 (Public Review):**
Summary:In this paper, Song, Shi, and Lin use an existing deep learning-based sequence model to derive a score for each haplotype within a genomic region, and then perform association tests between these scores and phenotypes of interest. The authors then perform some downstream analyses (fine-mapping, various enrichment analyses, and building polygenic scores) to ensure that these associations are meaningful. The authors find that their approach allows them to find additional associations, the associations have biologically interpretable enrichments in terms of tissues and pathways, and can slightly improve polygenic scores when combined with standard SNP-based PRS.Strengths:I found the central idea of the paper to be conceptually straightforward and an appealing way to use the power of sequence models in an association testing framework.The findings are largely biologically interpretable, and it seems like this could be a promising approach to boost power for some downstream applications.Weaknesses:The methods used to generate polygenic scores were difficult to follow. In particular, a fully connected neural network with linear activations predicting a single output should be equivalent to linear regression (all intermediate layers of the network can be collapsed using matrix-multiplication, so the output is just the inner product of the input with some vector). Using the last hidden layer of such a network for downstream tasks should also be equivalent to projecting the input down to a lower dimensional space with some essentially randomly chosen projection. As such, I am surprised that the neural network approach performs so well, and it would be nice if the authors could compare it to other linear approaches (e.g., LASSO or ridge regression for prediction; PCA or an auto-encoder for converting the input to a lower dimensional representation).

Response: We thank the reviewer for the recognition and valuable suggestion on our work. Just as the reviewer suggested, our polygenic prediction procedure is equivalent to linear transformation and in this revision, we indeed found that it was unnecessary to use neural network framework to replace linear model. Indeed, both our result and previous work indicated that linear model fitted polygenic traits better than non-linear one, which was also the reason we chose linear activation for neural network in the original manuscript.

In this revision, we followed the reviewer’s suggestion to apply a more straightforward linear framework for polygenic prediction. We first calculated weighted sum of HFS for each block (1,361 independent blocks in total), then, in each target ancestry, we used LASSO regression to integrate them with SNP PRS into one final score. We also conducted comparative analysis in British European test set and found that LASSO, ridge and elastic net gave similar result, and LASSO performed slightly better. By applying this straightforward framework and sliding window strategy, we moderately improved the prediction performance.

Line 349: “Using height as a representative trait, we first estimated the proportion of variance captured by top loci, and found that HFS of loci with PIP>0.4 (n=5,101) captured roughly 80% of variance explained by all genome-wide loci (n=1,200,024 corresponded to sling-window strategy; Figure 5A). We then calculated HFS+LDAK in non-British European (NBE), South Asian (SAS), East Asian (EAS) and African (AFR) population in UK Biobank, and observed 17.5%, 16.1%, 17.2% and 39.8% improvement over LDAK alone (p=3.21×10-16, 0.0001, 0.002 and 0.001, respectively. Figure 5C).”

A very interesting point of the paper was the low R^2 between the HFS scores in adjacent windows, but the explanation of this was unclear to me. Since the HFS scores are just deterministic functions of the SNPs, it feels like if the SNPs are in LD then the HFS scores should be and vice versa. It would be nice to compare the LD between adjacent windows to the average LD of pairs of SNPs from the two windows to see if this is driven by the fact that SNPs are being separated into windows, or if sei is somehow upweighting the importance of SNPs that are less linked to other SNPs (e.g., rare variants).

Response: We thank the reviewer for the suggestion on understanding LD mechanism. In this revision, we used chromosome 1 as an example and calculate the pairwise LD among all SNPs within two adjacent loci. As shown in Figure S1 (below), although HFS-based LD is still significantly lower than median SNP-based LD (paired Wilcoxon test p=1.76e-5), we found that median SNP LD between loci was still lower than what typically observed between adjacent SNPs in GWAS (histogram of x axis; median = 0.06). We reasoned that dividing SNPs into block is one of the reasons that HFS suffer less LD than standard GWAS, but not the whole story.

**Author response image 2. sa3fig2:** 

We agree with the reviewer that the effect of rare variants could also play an important role. In fact, sei author has also found that rare variants tended to have larger sei-predicted effects. We conducted an approximate analysis that remove all rare variants and repeated HFS calculation. Indeed, here HFS LD has profoundly raised to median=0.14, indicating that involving rare variants was vital for low LD.

**Author response image 3. sa3fig3:** 

Line 123: “Further evaluation indicated that this low LD was led by two factors: integration of rare variant impacts and segmentation. Firstly, excluding rare variants from HFS caused the LD raised to median=0.14 (Method; Figure S2C). Secondly, median LD of SNPs from adjacent loci was 0.06, which was significantly higher than HFS LD (paired Wilcoxon p=1.76×10-5) but significantly lower than HFS LD without rare variants (paired Wilcoxon p<2.2×10-16).”

There were also a number of robustness checks that would have been good to include in the paper. For instance, do the findings change if the windows are shifted? Do the findings change if the sequence is reverse-complemented?

Response: Following the reviewer’s suggestion, we conducted a sliding window analysis where all loci were shifted 2048 bp, thereby doubling the total number of loci. In fine-mapping analysis, more than 90% of the causal loci were reproduced in sliding window analysis, either by themselves or by a overlapping locus:

Line 207: “29.4% of causal loci (PIP>0.95) in the original analysis were still causal in sliding window analysis. 31.1% and 29.3% of causal loci whose 5’ and 3’ overlapping locus had PIP>0.95 in sliding window analysis, respectively, while themselves were no longer causal.”

In polygenic prediction analysis, sliding window strategy significantly improved prediction accuracy, as we discussed in question 1.

As for the issue of reverse complement, the nature of sei input layer is to encode both strand in a symmetric manner, such that the output for both strands would be the same. We have also run sei on the reverse complement (generated by seqkit seq -r -p) to verify that original sequence and reverse complement give the same output.

It was also difficult to contextualize the present work in terms of recent results showing that sequence models tend to not do very well at predicting cross-individual expression changes (and such results presumably hold for predicting cross-individual chromatin changes). In particular, it would be good for the authors to contrast their findings with the work of Alex Sasse and colleagues (https://www.biorxiv.org/content/10.1101/2023.03.16.532969.abstract) and Connie Huang and colleagues (https://www.biorxiv.org/content/10.1101/2023.06.30.547100.abstract).

Response: Following the reviewer’s suggestion, we added a new discussion paragraph on the issue of sequence model performance on interindividual variations. In brief, we suggest that although the drawback of lack of cross-individual training sets exists and future improvement is necessary, chromatin changes could be better predicted than gene expression. This is because the latter task requires information on long range interaction, which varies among genes and are difficult to be captured by using reference genome as training set. We made a schematic to clarify this:

**Author response image 4. sa3fig4:** 

We also noticed a few recent studies that directly validated sei predictions by experiments and showed significant accuracy, such as https://doi.org/10.1016/j.neuron.2022.12.026. Taken together, while we agreed that it is necessary to improve sequence model by adding more cross-individual training samples, the current SOTA model sei could still provide unique value to our study.

Line 423: “The challenge of using sequence-based deep learning (DL) models in HFS applications is further compounded by their difficulty in predicting variations between individuals. Recent studies(Huang et al., 2023; Sasse et al., 2023) indicate that DL models, trained on the reference human genome, demonstrate limited accuracy in predicting gene expression levels across different individuals. This limitation is likely due to the models' inability to account for long-range regulatory patterns, which are crucial for understanding the impact of variants on gene expression and vary across genes. In contrast, our study leveraged sequence-determined functional genomic profiles in association studies, which mitigates this issue to an extent. For instance, although sei cannot identify the specific gene regulated by a given input sequence, it can predict changes in the sequence's functional activity. Future improvements in DL models' ability to predict interindividual differences could be achieved by incorporating cross-individual data in the training process. An example of such data is the EN-TEX(Rozowsky et al., 2023) dataset, which aligns functional genomic peaks with the specific individuals and haplotypes they correspond to.”

**Reviewer #2 (Public Review):**
Summary:In this work, Song et al. propose a locus-based framework for performing GWAS and related downstream analyses including finemapping and polygenic risk score (PRS) estimation. GWAS are not sufficiently powered to detect phenotype associations with low-frequency variants. To overcome this limitation, the manuscript proposes a method to aggregate variant impacts on chromatin and transcription across a 4096 base pair (bp) loci in the form of a haplotype function score (HFS). At each locus, an association is computed between the HFS and trait. Computing associations at the level of imputed functional genomic scores should enable the integration of information across variants spanning the allele frequency spectrum and bolster the power of GWAS.The HFS for each locus is derived from a sequence-based predictive model. Sei. Sei predicts 21,907 chromatin and TF binding tracks, which can be projected onto 40 pre-defined sequence classes ( representing promoters, enhancers, etc.). For each 4096 bp haplotype in their UKB cohort, the proposed method uses the Sei sequence class scores to derive the haplotype function score (HFS). The authors apply their method to 14 polygenic traits, identifying ~16,500 HFS-trait associations. They finemap these trait-associated loci with SuSie, as well as perform target gene/pathway discovery and PRS estimation.Strengths:Sequence-based deep learning predictors of chromatin status and TF binding have become increasingly accurate over the past few years. Imputing aggregated variant impact using Sei, and then performing an HFS-trait association is, therefore, an interesting approach to bolster power in GWAS discovery. The manuscript demonstrates that associations can be identified at the level of an aggregated functional score. The finemapping and pathway identification analyses suggest that HFS-based associations identify relevant causal pathways and genes from an association study. Identifying associations at the level of functional genomics increases the portability of PRSs across populations. Imputing functional genomic predictions using a sequence-based deep learning model does not suffer from the limitation of TWAS where gene expression is imputed from a limited-size reference panel such as GTEx.However, there are several major limitations that need to be addressed.Major concerns/weaknesses:(1) There is limited characterization of the locus-level associations to SNP-level associations. How does the set of HFS-based associations differ from SNP-level associations?

Response: We thank the reviewer for the recognition and the valuable suggestion on our manuscript. Following the reviewer’s suggestion, in this revision we added a paragraph to compare the basic characteristics between HFS-based and SNP-based association study. These comparisons suggested that HFS had no advantage in testing marginal association, but performed better in detecting causal associations.

Line 144: “When comparing HFS association with the standard SNP-based GWAS on the same data, we found that 98% of significant HFS loci also harbored a significant SNP. There were a few cases (n=0~5) where significant HFS loci did not harbored even marginal SNP association (GWAS p>0.01), which were due to the lack of common SNP in these loci. HFS association p value was higher than GWAS p value in 95 % of significant loci, suggested that HFS did not improve power to detect marginal effect. The genomic control inflation factor (λGC) for the HFS association test varied between 0.99 for asthma and 1.50 for height, closely resembling the SNP GWAS (Pearson Correlation Coefficient [PCC]=0.91, paired t-test p=0.16; Method and Figure S3). We concluded that HFS-based association tests had adequate power and do not introduce additional p-value inflation.”

(2) A clear advantage of performing HFS-trait associations is that the HFS score is imputed by considering variants across the allele frequency spectrum. However, no evidence is provided demonstrating that rare variants contribute to associations derived by the model. Similarly, do the authors find evidence that allelic heterogeneity is leveraged by the HFS-based association model? It would be useful to do simulations here to characterize the model behavior in the presence of trait-associated rare variants.

Response: Following the reviewer’s suggestion, we conducted a sensitivity analysis that removed all rare (MAF<0.01) variants and repeated the HFS analysis (HFScommon) on chromosome 1. In linear association analysis, we found that 10.6% of HFS signals (p<5×10-8) were missed by HFScommon. In fine-mapping, 55.3% of HFS causal signals (PIP>0.95) were missed by HFScommon. We concluded that rare variants played an important role in the performance of HFS, especially its advantages in fine-mapping.

Line 175: “We also found that rare variants played an important role in the good find-mapping performance of HFS: when variants with MAF<0.01 were removed, 55.3% of the causal signals would be missed in HFS+SUSIE analysis.”

We then attempted to conduct a simulation analysis where rare variants were causal to the phenotype, and the association statistics were the same as real GWAS of height. However, such simulation seemed not to properly reflect real scenario: no matter how we changed the association between rare variants and the phenotype, HFS association p-value could hardly reached the significance level of SNP association. We proposed that this is because simulation could not properly reflect how variants impact functional genomics: in fact, when randomly selected a rare variant as causal variant, there is high possibility that it had no impact on functional genomics, therefore its HFS would be close to zero. When such a variant was set as causal (which is unlikely in real scenario), HFS would not properly capture the association. We reasoned that it might be difficult to evaluate HFS by simulation, since the nonlinear relation between SNP and HFS as well as among SNPs were difficult to be properly simulated.

**Author response image 5. sa3fig5:** 

(3) Sei predicts chromatin status / ChIP-seq peaks in the center of a 4kb region. It would therefore be more relevant to predict HFS using overlapping sequence windows that tile the genome as opposed to using non-overlapping windows for computing HFS scores. Specifically, in line 482, the authors state that "the HFS score represents overall activity of the entire sequence, not only the few bp at the center", but this would not hold given that Sei is predicting activity at the center for any sequence.

Response: We thank the reviewer for the suggestion on sliding window design. In this revision, we shifted all loci 2,048 bp to double the number of loci and repeated the fine-mapping and polygenic prediction analysis. For fine-mapping, we found that the result was generally robust with regard to sliding window procedure, and the majority of the causal associations were retained:

Line 207: “29.4% of causal loci (PIP>0.95) in the original analysis were still causal in sliding window analysis. 31.1% and 29.3% of causal loci whose 5’ and 3’ overlapping locus had PIP>0.95 in sliding window analysis, respectively, while themselves were no longer causal.”

In polygenic prediction, sliding window analysis provided a significantly improved performance compared with previous analysis on non-overlapping loci:

However, since in this revision we have several updates on the polygenic prediction procedure, it was difficult to quantify how much improvement was led by sliding window design. Thus, we directly showed the new result in figure 5 but did not compare it with the original result.

We also modified the previously imprecise statement to:

Line 490: “…it integrated information of the entire sequence, not only the few bp at the center.”

(4) Is the HFS-based association going to miss coding variation and several regulatory variants such as splicing variants? There are also going to be cases where there's an association driven by a variant that is correlated with a Sei prediction in a neighboring window. These would represent false positives for the method, it would be useful to identify or characterize these cases.

Response: As the reviewer suggested, sei captured only functional genomic features and is by nature prone not to perform well when the causal variants impact protein sequences. In this revision, we characterized this by focusing on causal exonic variants (SNP PIP>0.95):

Line 322: “On the other hand, HFS perform worse than SNP-based fine-mapping on exonic regions. Taking height as an example, PolyFun detected 125 causal SNPs (PIP>0.95) in the exonic regions, but only 16% (20) of loci that harbored them also reached PIP>0. 5 (11 reached PIP>0.95) in HFS+SUSIE analysis. Among the 105 loci that missed such signals (HFS PIP<0.5), 12 had a nearby locus (within 10kb) showing HFS PIP>0.95, which likely reflected false positive led by LD. Thus, SNP-based analysis should be prioritized over HFS in coding regions.”

Additional minor concerns:(1) It's not clear whether SuSie-based finemapping is appropriate at the locus level, when there is limited LD between neighboring HFS bins. How does the choice of the number of causal loci and the size of the segment being finemapped affect the results and is SuSie a good fit in this scenario?

Response: Following the reviewer’s suggestion, we reran SUSIE under different predefined causal loci number (from 2 to 10), and found that the identified causal loci were consistent.

**Author response image 6. sa3fig6:** 

Line 211: “Besides, HFS+SUSIE was also robust when the predefined number of causal loci (L=2 to 10) was changed, and the number of detected loci were not changed.”

As for the size of segmentation, we divided the predefined segmentations (independent blocks detected by LDetect) into two half and reran SUSIE, and found that three additional causal loci emerged in one half. This suggested that using too small segmentation might increase the false positive rate. However, since there is no LD between independent blocks (which was guaranteed by LDetect), it is not necessary to use even longer blocks.

**Author response image 7. sa3fig7:** 

Line 133: “Simulation analysis revealed that when a non-reference sequence class score was associated the trait, reference class score could still capture median 70% of HFS-trait association R2.”

(2) It is not clear how a single score is chosen from the 117 values predicted by Sei for each locus. SuSie is run assuming a single causal signal per locus, an assumption which may not hold at ~4kb resolution (several classes could be associated with the trait of interest). It's not clear whether SuSie, run in this parameter setting, is a good choice for variable selection here.

Response: As we discussed below (question 3), in this revision we no longer applied SUSIE to find one sequence class score for each locus due to the impact of overfitting, and use the reference sequence class uniformly for all loci. As reviewer suggested, we applied simulation to evaluate how this procedure influence HFS performance, especially when multiple sequence class of the same locus is causal to the phenotype. We found that reference sequence class score could capture median 69.1% of phenotypic R2 when the causal sequence class is not the reference, and captured median 59.2% of R2 when there was 2~5 non-reference causal class. We concluded that the loss led by skipping sequence class selection is mild, and it is necessary to do so in consideration of the risk of overfitting.

**Author response image 8. sa3fig8:** 

(3) A single HFS score is being chosen from amongst multiple tracks at each locus independently. Does this require additional multiple-hypothesis correction?

Response: We agree with the reviewer that choosing the sequence class for each locus represented multiple testing, and with additional experiments we indeed observed some evidences of overfitting of this procedure. Thus, in this revision, we no longer applied the per-locus feature selection procedure, but instead used the sequence class corresponded to the reference (hg38) sequence. Consequently, additional multiple-testing correction is avoided with this procedure. We admitted that such simplification missed certain information, but as mentioned above, such lost is moderate, and is necessary to ensure statistical robustness and reduce false positive. In fact, with such simplification we better controlled the inflation factor of HFS GWAS and got better portability in polygenic prediction.

(4) The results show that a larger number of loci are identified with HFS-based finemapping & that causal loci are enriched for causal SNPs. However, it is not clear how the number of causal loci should relate to the number of SNPs. It would be really nice to see examples of cases where a previously unresolved association is resolved when using HFS-based GWAS + finemapping.

Response: In this revision, we did not observe a clear relation between causal loci number and causal gene number. The only trend is that SNP-based fine-mapping seemed to perform better at coding regions, in accordance with the fact that HFS capture functional genomic signals.We also added new interpretations to highlight some examples where HFS resolve previously unresolved association signals. For example,

Line 287: “Specifically, in 1q32.1 region, HFS+SUSIE identified two loci with PIP>0.9 (Figure 4B). SNP-based association also found significant association in this region, but SNP fine-mapping(Weissbrod et al., 2020) could not resolve this signal and only found seven signals between PIP=0.1 to 0.5.”

(5) Sequence-based deep learning model predictions can be miscalibrated for insertions and deletions (INDELs) as compared to SNPs. Scaling INDEL predictions would likely improve the downstream modeling.

Response: Following the reviewer’s suggestion, we conducted a sensitivity analysis that removed all indel on chromosome 1 and repeated HFS analysis. Removing indel has indeed increased the number of significant (p<5e-8) association by 9%, but also slightly increased inflation factor (paired wilcox test p=0.0001). In fine mapping analysis, removing indel caused a 4.7% decrement in the number of detected causal association (PIP>0.95). We reasoned that the potential miscalibration on indel has indeed impacted the statistical power of HFS, but the proper approach to control this impact might not be direct and is still await optimizing. In this revision, we still kept all indels in the analysis, since we proposed that the power of fine-mapping is more important than the power of marginal association.

Line 213: “Lastly, removing insertion and deletion would reveal 9% more significant association (p<5×10-8) but 4.7% less causal association (PIP>0.95), and slightly increased inflation factor (Wilcoxon p=0.0001, Figure S4).”

**Author response image 9. sa3fig9:** 

**Reviewer #1 (Recommendations For The Authors):**
It was unclear to me why the sei output was rounded to two decimal places to "avoid influence of sei prediction noise". Wouldn't rounding introduce additional noise?

Response: We thank the reviewer for pointing out our inadequate description. The rounding procedure is used to mask the low value that likely did not reflect any real change. The idea is that, even if a variant actually does not bring about any functional changes, sei would still output a very low HFS value that is not equal to, but close to, zero. By rounding procedure, such low values would be set to zero, which could avoid noise. We have added this rationale to the method section:

Line 529: “This is due to the fact that even if a variant actually makes no impact on functional genomics, sei would still output a value that are close to but not equal to reference sequence class score. Rounding procedure would set such HFS to zero and remove the random value from sei.”

Minor comments / typos:There are many typos in the abstract.

Response: We have revised the typo and grammar issues in the abstract in this revision.

I believe "Arachnoid acid-intelligence" should be "Arachidonic acid-intelligence".Consistently there is no space between text and parenthetical citations. For example, "sei(Chen et al., 2022)" should be "sei (Chen et al., 2022)".Line 110: "at least one non-reference haplotypes"  "at least one non-reference haplotype".Line 155: "data-based method"  "data-based methods".Lines 165-166: "functionally importance"  "functional importance".

Response: We have made these revisions accordingly.

Line 210: the sentence containing "this annotation on conditioned of a set of baseline annotations" is unclear.

Response: We have revised this sentence as “…regressed the PIP against this annotation, with a set of baseline annotations included as covariates, similar to the LDSC framework.”

Line 213: "association"  "associations".Line 219: "association"  "associations".Line 251: "result"  "results".Line 269: "result"  "results".Line 289: "known to involved"  "known to be involved".Line 356: "LDAK along"  "LDAK alone".Line 362: "BOLT-LMM along"  "BOLT-LMM alone".Supplement: "Hihglighted"  "Highlighted".

Response: We have made these revisions accordingly.

Line 444: Were "British ancestry Caucasians" defined as individuals that self-identified as "white British"? If so, then they should be described as "self-identified "white British"".

Response: As the reviewer pointed out, we have changed the description as self-identified British ancestry Caucasians.

**Reviewer #2 (Recommendations For The Authors):**
(1) A 2022 cistrome-wide association study (CWAS) computed associations between genetically-predicted chromatin activity and phenotypes. Adding a reference to this paper would be helpful. https://pubmed.ncbi.nlm.nih.gov/36071171/

Response: Following the reviewer’s suggestion, we discussed the similarity between CWAS and our study:

Line 89: “In line with this notion, a recent similar strategy called cistrome-wide association study (CWAS) integrated variant-chromatin activity and variant-phenotype association to boost power of genetic study of cancer. (Baca et al., 2022).”

(2) Line 487 states: "We applied sei to predict 21,906 functional genomic tracks for each sequence, without normalizing for histone mark." It's not clear what normalization is being referred to here.

Response: We have revised the sentence to:

Line 495: “We applied sei to predict 21,906 functional genomic tracks for each sequence, without normalizing for histone mark (divided each track score by the sum of histone mark score) as suggested by the sei author.”

(3) The figures are extremely low resolution, they need to be updated.

Response: In this revision, we uploaded separate pdf file for each figure to provide high resolution graphs.

(4). The results section was difficult to follow and would benefit from being written more clearly.

Response: In this revision, we re-arranged some of the result section to better clarify the main idea. We moved all statistical results to the bracket and focused our main text on the interpretation. For example,

Line 123: “Further evaluation indicated that this low LD was led by two factors: integration of rare variant impacts and segmentation. Firstly, excluding rare variants from HFS caused the LD raised to median=0.14 (Method; Figure S2C). Secondly, median LD of SNPs from adjacent loci was 0.06, which was significantly higher than HFS LD (paired Wilcoxon p=1.76×10-5) but significantly lower than HFS LD without rare variants (paired Wilcoxon p<2.2×10-16).”

(5) "Along" is used several times in the final results section (PRS estimation), this should be "alone".

Response: We have modified all misused “along” by “alone” in this revision.

(6) Instead of using notation identifying genomic location, it might be clearer to provide gene names when illustrating examples of trait-associated promoters.

Response: In this revision, we added gene name of the corresponding promoters to the main text to better clarify the findings.